# Studies on Morphophysiological and Biochemical Parameters for Sucking Pest Tolerance in Organic Cotton

Shradha S. Aherkar [1], Surendra B. Deshmukh [2], Nitin. M. Konde [3], Aadinath N. Paslawar [4], Tanay Joshi [5,*], Monika M. Messmer [6] and Amritbir Riar [5]

[1] Department of Agricultural Botany, Dr. Panjabrao Deshmukh Krishi Vidyapeeth, Akola 444104, India; shradhaaherkar@gmail.com

[2] Cotton Research Unit, Dr. Panjabrao Deshmukh Krishi Vidyapeeth, Akola 444104, India; suren.pdkv@gmail.com

[3] Department of Soil Science and Agricultural Chemistry, Dr. Panjabrao Deshmukh Krishi Vidyapeeth, Akola 444104, India; nitinkonde75@gmail.com

[4] Department of Agronomy, Dr. Panjabrao Deshmukh Krishi Vidyapeeth, Akola 444104, India; adinathpaslawar@rediffmail.com

[5] Group Resilient Cropping System, Department of International Cooperation, Research Institute of Organic Agriculture (FiBL), 5070 Frick, Switzerland; amritbir.riar@fibl.org

[6] Group Plant Breeding, Department of Crop Science, Research Institute of Organic Agriculture (FiBL), 5070 Frick, Switzerland; monika.messmer@fibl.org

* Correspondence: tanay.joshi@fibl.org

**Abstract:** The demand for organic cotton is primarily driven by manufacturers and brands with a corporate focus on environmental and social responsibility. These entities strive to be responsible stewards by seeking organic cotton, which not only offers environmental benefits but also provides softer, more durable, and longer-lasting clothing. Unlike conventional cotton, organic cotton is processed without the use of harsh chemicals, making it more comfortable for individuals with sensitive skin. A study was conducted at the Center of Organic Agriculture Research and Training Center, Department of Agronomy, Dr. PDKV, Akola, Maharashtra, India to evaluate 22 cotton genotypes, including control samples, using a randomized block design with three replications during the kharif (June–October) season in the years 2019–2020 and 2020–2021, under complete organic conditions. During the initial year of the study (2019–2020), visual observations were made to assess the incidence of sucking pests on the cotton genotypes' leaves, including the top, middle, and bottom portions. The observations indicated promising results, leading to a more detailed study in the subsequent year (2020–2021). This extended study identified several tolerant genotypes to sucking pests, such as AV-G11, PA-255, GA-8004, AV-C14, and AV-G13 from the *arboreum* species, as well as AKH-09-5, a *hirsutum* cultivar. Among the recorded data, it was found that the lowest mean aphid population occurred at 90 days after sowing (DAS), with only 1.53 aphids per leaf. Similarly, the lowest mean populations of Cicadellidae, thrips, and whitefly were recorded at 0.75, 0.97, and 0.63 per leaf, respectively, all at 30 DAS. Microscopic analysis of trichome density and gossypol glands revealed a negative and significant correlation with *Aphis gossypi* (aphids), *Cicadellidae* sp. (*Cicadellidae*), and *Thrips tabaci* (thrips). However, a positive and significant correlation was found with *Bemesia tabaci* (whitefly). Furthermore, the estimation of total soluble sugar using the Anthrone method, total nitrogen, and crude protein showed positive and significant correlations with aphids, *Cicadellidae*, and thrips, but negative, significant correlations with whitefly. The findings indicate that higher trichome density provides greater resistance to sucking pest infestation. It was concluded that *G. arboreum* genotypes exhibit greater tolerance to sucking pests compared to *G. hirsutum* varieties. This implies that *G. arboreum* varieties may require less intensive pest management, aligning with organic farming principles. The discovery of these genotypes opens up possibilities for utilizing them as sustainable and pest-resistant options in cotton cultivation, promoting environmentally friendly and organic farming practices in cotton fiber production.

**Keywords:** organic; cotton; sucking pests; trichome density; gossypol glands; total sugar; total nitrogen; crude protein

## 1. Introduction

India has been recognized as the "Cradle of the cotton industry" since ages. Recognized as the "King of Fibers" cotton is the most important fibre and cash crop. This white gold belonging to the genus *Gossypium* in the family *Malvaceae* has four cultivated species, viz. *Gossypium arboreum*, *G. herbaceum*, *G. hirsutum* and *G. barbadense*, occupying a place of pride in Indian agriculture and economics by earning valuable foreign exchange besides providing employment to millions of people [1,2].

Cotton was cultivated organically until the advent of chemical fertilizers and pesticides. Insect pest infestation was a major constrain in cotton production. Insecticides, herbicides, fungicides and chemical fertilizers were liberally applied to the cotton crop with the sole intention of reaping a richer harvest during the green revolution. Although, spraying insecticides effectively controlled the insect-pests but increased the cost of production, as well as pesticidal residue, which is not economically profitable and also dangerous for human health [3]. In order to achieve sustainable yields, there is an increasing need for eco-friendly pest control methods. The natural resources can be utilized optimally without harming the ecosystem, as well as disease, and pest control can be managed through targeted support of predators and parasitoids along with the cultivation of resistant varieties instead of using pesticides [4].

It is crucial to recognize the environmental and social impacts of our choices [5]. By transitioning to organic cotton fabrics, we can contribute to reducing the ecological footprint associated with conventional cotton production, which often involves intensive pesticide use and adverse effects on ecosystems. Organic cotton is widely acknowledged for is softness, recognized as hypoallergenic, and lasts for a longer time, compared to conventional cotton. The most important benefit of organic cotton is it causes lesser harm to the environment, reduces water wastage, is safe for livestock and ensures a safer working environment for farmers. The shift towards organic cotton will aid in promoting sustainable agricultural practices, safe-guarding biodiversity, and fostering a healthier environment for all living organisms. This shift also demands the capacities of extension services that need to be enhanced with appropriate training aimed at bridging the knowledge gap on the optimal use of resources, along with sustainable farming practices [6]. According to the WWF (World Wildlife Fund), 2700 litres of water is utilized to produce normal cotton needed to make just one T-shirt. However, organic cotton utilizes less water. The Soil Association says it uses only 243 litres of water by comparison [7]. As a result, an increasing number of fashion brands are supporting the natural fibre. The Textiles Exchange reported that organic cotton uses 91 per cent less 'blue' water, i.e., from groundwater and surface-water bodies, such as freshwater lakes and rivers, than conventional cotton [8]. However, less than one per cent of all cotton produced is currently organic, meaning there is huge potential for improvement when it comes to how we make the fibre suitable for industrial purposes. Hence, studies of genotypes, which perform better in organic conditions, have been performed so that efficient resistant cultivars can be evaluated for the further study of fibre properties, and may be beneficial for future research.

The objective of this study was to identify prominent and tolerant genotypes to various biotic stresses, particularly sucking pests, under organic cultivation conditions. For this study, we explored two species of cotton, and hypothesized that (1) *G. arboreum* being native species to Indian subcontinent can be more tolerant to cotton pest, and (2) sucking pest incidence is closely associated with different morpho-physiological, morphological and biochemical parameters.

## 2. Material and Methods

The study was carried out at Centre for Organic Agriculture Research and Training (COART), Dr. Panjabrao Deshmukh Krishi Vidyapeeth, Akola, Maharashtra, India during *Kharif* season (monsoon sowing, June–October) between 2019–2020 and 2020–2021, under organic conditions. Twenty cotton genotypes (Table 1), along with one resistant check DHY-286 and one susceptible check DCH-32, were grown in a Randomized Block Design in three replications. The total plot size was $1.8 \times 6.0$ m and the seeds were dibbled with the spacing of varieties $90 \times 30$ cm$^2$ and for hybrids $90 \times 60$ cm$^2$. All packages of practices for organic cotton cultivation were followed, except plant protection measures to study the tolerant genotype. The trial was grown under unprotected conditions for insect–pest infestation in both years. The observations were recorded on five randomly selected plants of each genotype from each replication on various sucking pest incidence in both years, whereas morpho-physiological, biochemical and morphological observations were recorded in the year 2020–2021 at 45, 60, 90 and 120 DAS. (Figure 1).

**Table 1.** List of genotypes from different cotton species evaluated in the study.

| Sr. No. | Cultivar No. | Genotypes | Type of Genotype |
|---------|--------------|-----------|------------------|
| 1 | SGF_001 | Namaskar Gold_81 | AH |
| 2 | SGF_002 | AV-G13 | AV |
| 3 | SGF_003 | AV-G11 | AV |
| 4 | SGF_006 | PA-255 | AV |
| 5 | SGF_008 | AV-C14 | AV |
| 6 | SGF_010 | GA-8004 | AV |
| 7 | SGF_301 | Suraj | HV |
| 8 | SGF_303 | Chetna_J1 | HV |
| 9 | SGF_305 | Chetna_D11 | HV |
| 10 | SGF_306 | Chetna_S1 | HV |
| 11 | SGF_318 | GH-8032 | HV |
| 12 | SGF_319 | GH-8001E | HV |
| 13 | SGF_321 | BKMP_27 | HV |
| 14 | SGF_322 | AKH09-5 | HV |
| 15 | SGF_323 | AKH-9916 | HV |
| 16 | SGF_701 | Mallika | HH |
| 17 | SGF_705 | Nirmal996 | HH |
| 18 | SGF_723 | Vasudha-1318 Gold | HH |
| 19 | SGF_718 | Bhakti-245 | HH |
| 20 | SGF_719 | Raja-954 | HH |
| 21 | | DHY-286 (R. check) | HV |
| 22 | | DCH-32 (S. check) | HH |

Note: SGF—Seeding the Green Future; AH—*arboreum* hybrid; AV—*arboreum* variety; HH—*hirsutum* hybrid; HV—*hirsutum* variety.

### 2.1. Entomological Study

The sucking pest, such as aphid (*Aphis gossypi*), leafhopper (*Cicadellid* sp.), thrips (*Thrips tabaci*) and whitefly (*Bemisia tabaci*), were observed periodically at 45, 60 and 90 DAS (Days after sowing) on the top, middle and bottom leaves of five randomly selected plants of each genotype in each replication. (Figure 2).

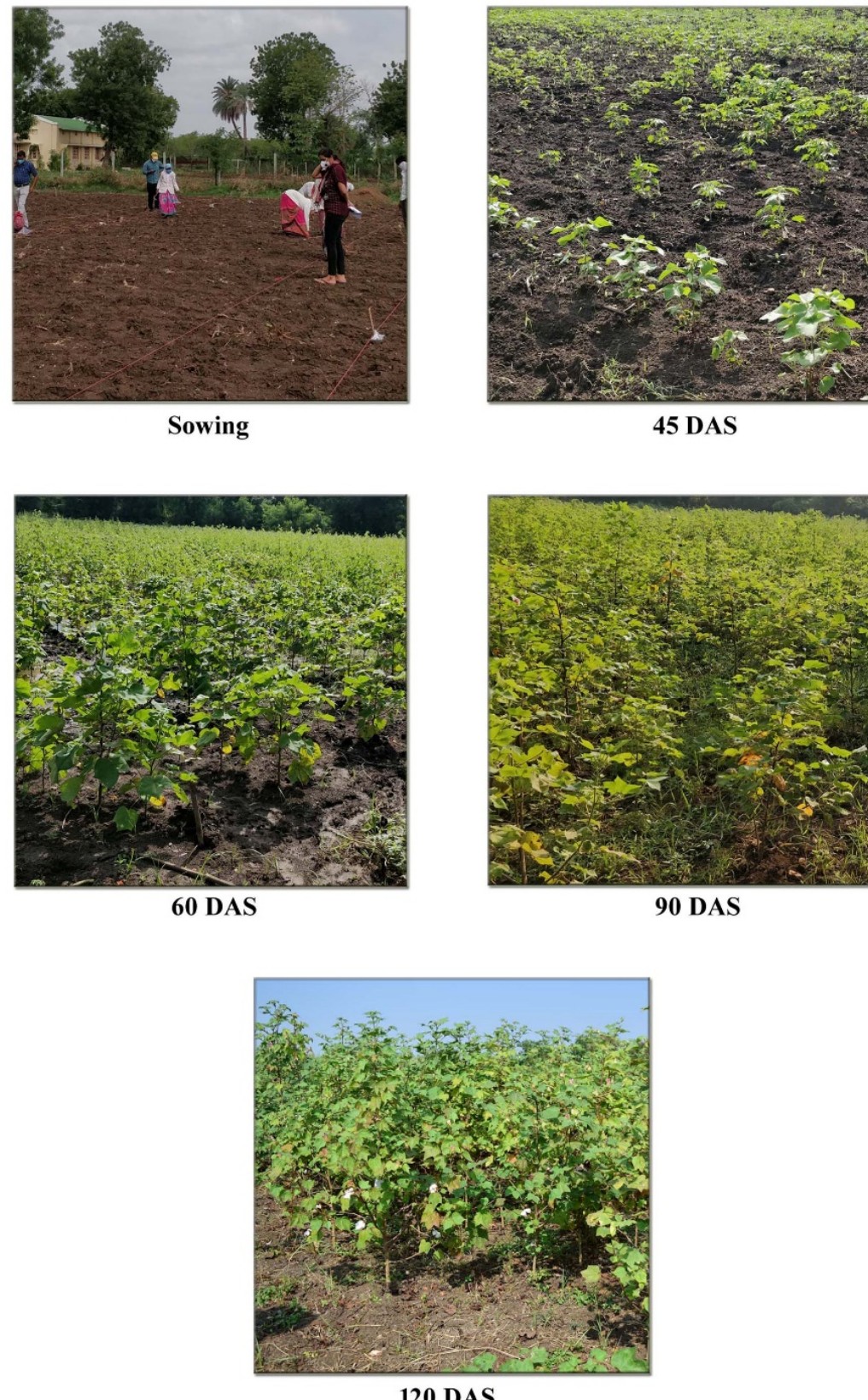

**Figure 1.** Different stages of crop growth for observation.

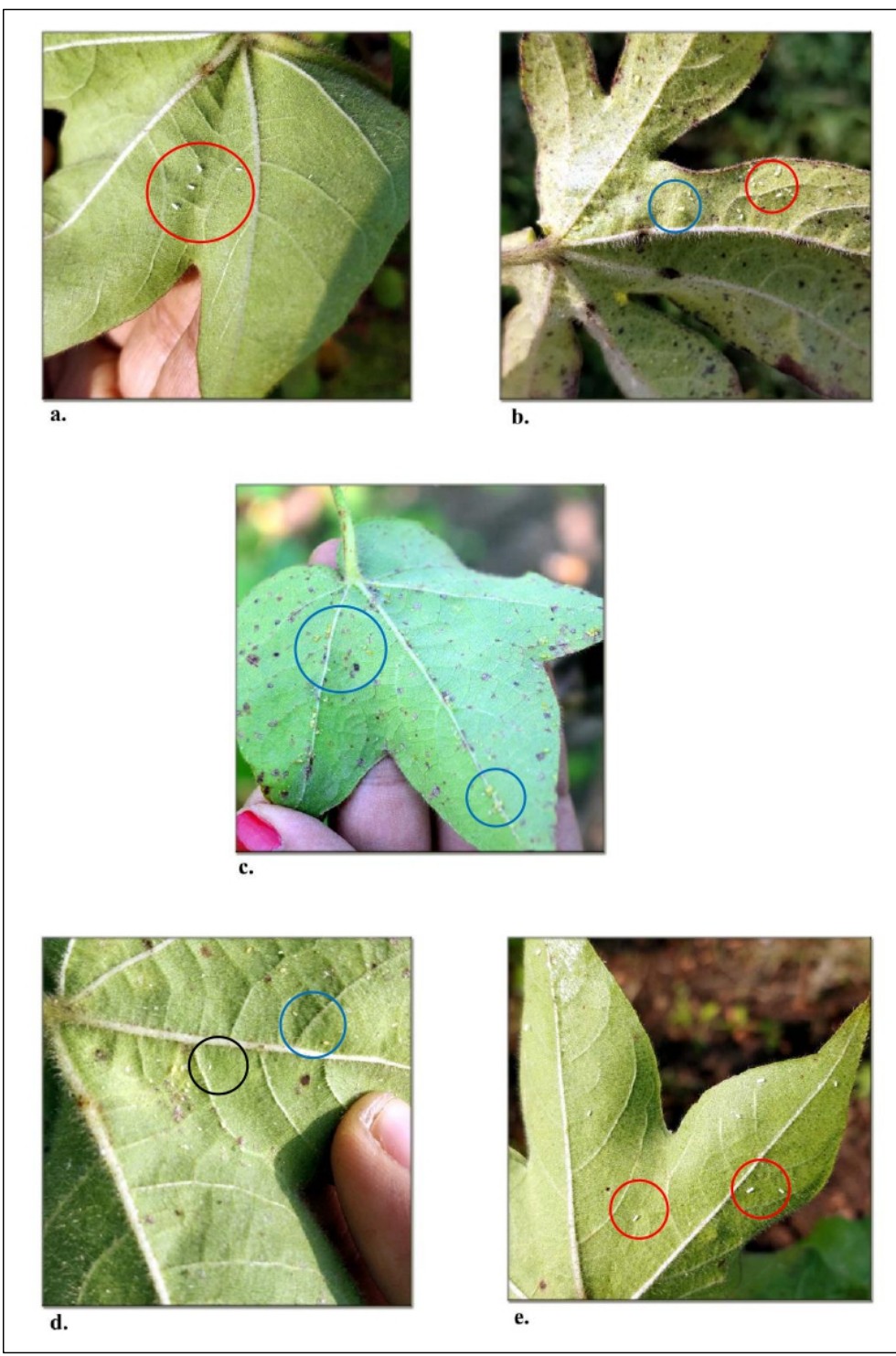

**Figure 2.** Sucking pest infestation on the leaves of cotton genotypes. Note: (**a**,**e**) Whiteflies sitting on the backside of the cotton leaf are clearly visible in the red circle; (**b**) the blue circle indicates the aphid population and the whitefly population in the red circle; (**c**) shows heavy infestation of aphid on the lower side of the leaf, indicated by the blue circle; (**d**) the blue circle indicates the aphid population and the black circle indicates the thrips population.

### 2.2. Morpho-Physiological Observation

The morpho-physiological characters, such as gossypol glands per cm$^2$, trichome density per cm$^2$ and chlorophyll content index, were observed at the same time interval as that of the entomological study.

The chlorophyll content index was measured with help of SPAD 2.0 m, where the leaf was pressed in middle of the SPAD meter and digital readings were obtained on field; thus, observations were recorded (Figure 3).

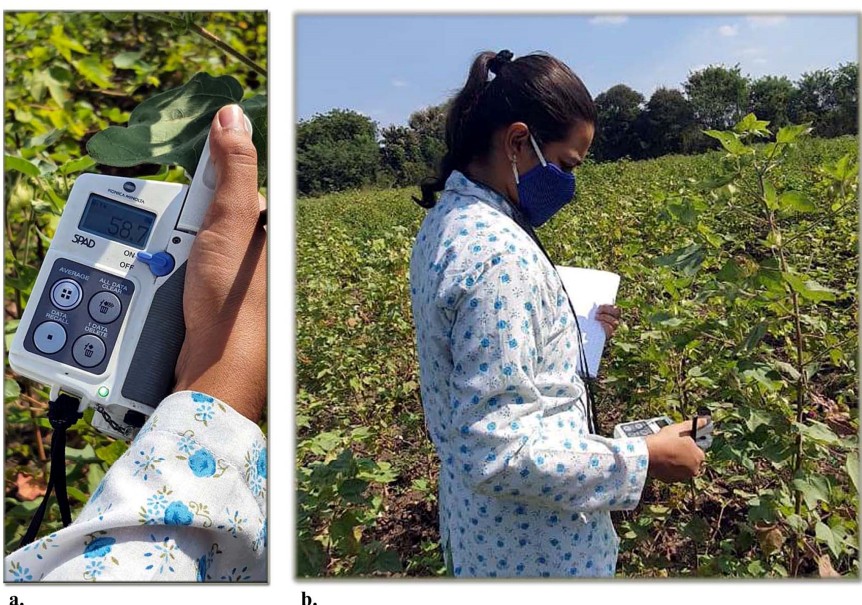

**a.**    **b.**

**Figure 3.** Use of SPAD 2.0 meter for measuring chlorophyll content. Note: (**a**) The digital SPAD 2.0 meter displaying the chlorophyll content of 58.7 in the cotton leaf. (**b**) Observations being taken on the field with the use of SPAD 2.0 meter at 60 DAS (days after sowing).

Gossypol glands were observed by cutting the leaf into a one-centimeter section and observing it under a compound microscope; the number of gossypol glands were counted (Figure 4).

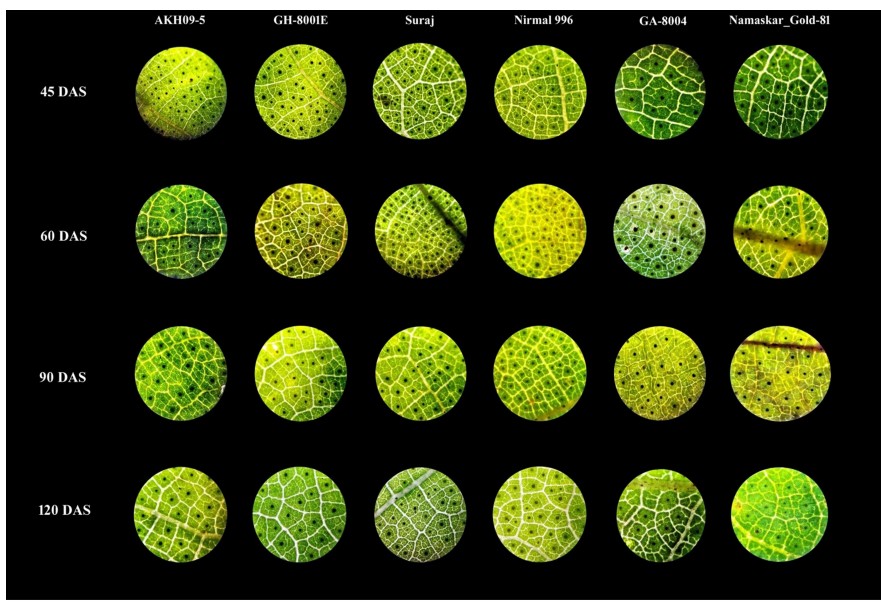

**Figure 4.** Gossypol glands observed at various days interval on various cotton genotypes under microscope.

Trichome density per cm$^2$ was recorded for which the leaf section was mounted on a slide in a drop of lactic acid and observed under the microscope at 10× magnification and observations were recorded (Figure 5).

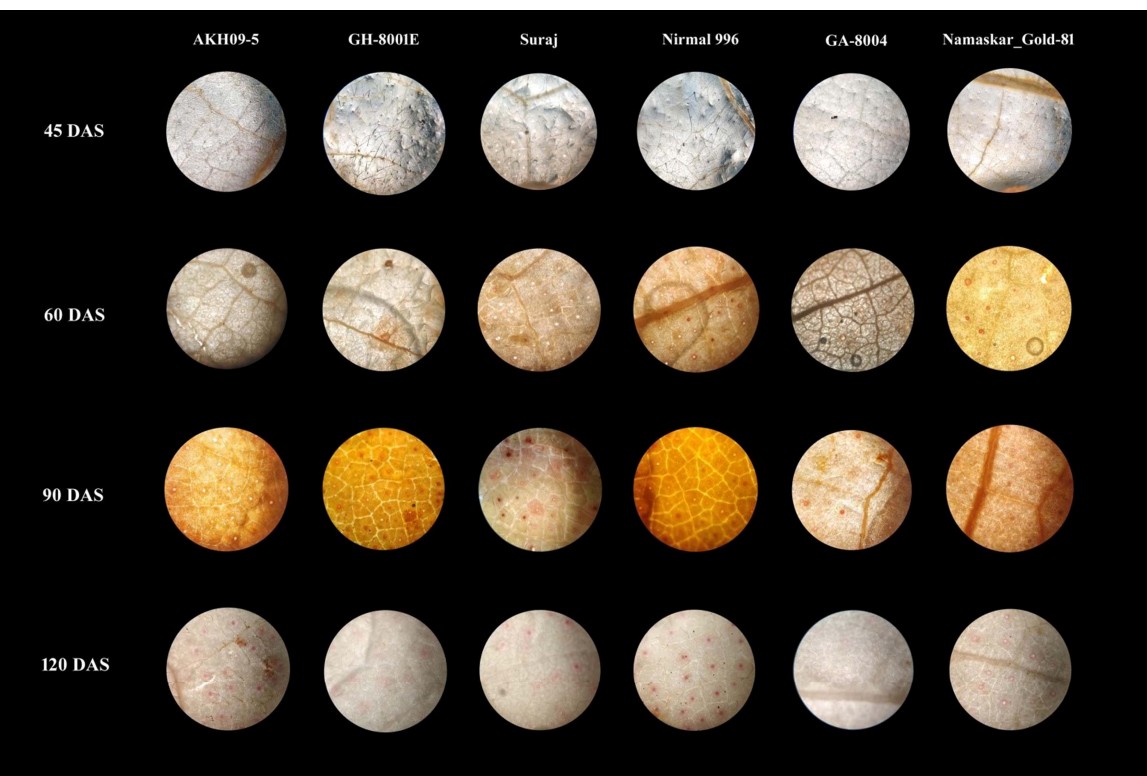

**Figure 5.** Trichomes (hair like structure) observed on the abaxial side of cotton leaves under microscope.

### 2.3. Biochemical Observations

The biochemical characters such as total soluble sugar was estimated by Anthrone's method [9], and total nitrogen as well as crude protein was estimated by the Kjeldhal method [10]. All the above-mentioned observations were recorded at 45, 60, 90 and 120 DAS (Table 2).

2.3.1. Total Soluble Sugar (Anthrone Method)

Leaf samples from the randomly selected plants of each genotype were collected at 45, 60 and 90 DAS. The leaf samples were washed thoroughly with distilled water and were kept on the hot water bath in 2.5 N HCl solution for preparation of the extract and then cooled to room temperature. Sodium carbonate was added to neutralize the solution and cease the effervescence. Distilled water was added to make up the final volume to 100 mL and then the solution was centrifuged. The supernatant was collected and 0.5 mL aliquot was taken in a separate flask and distilled water was added. The anthrone reagent was added to the aliquot and kept on the boiling hot-water bath. The aliquot turned green in color; it was then cooled to room temperature and with the help of a spectrophotometer, the absorbance was recorded at 630 nm for each sample. Simultaneously, standards were prepared and their absorbances were also recorded at 630 nm. The calculations were carried out using the given formula:

$$\text{Concentration of Unknown Sample (Total Soluble Sugar)}\left(\frac{\text{mg}}{\text{mL}}\right) = \frac{(A) \times (Y)}{(X)} \times 100 \quad (1)$$

where A = Concentration of Standard sample; X = Absorbance of Standard sample; and Y = Absorbance of unknown sample.

**Table 2.** Morphophysiological and Biochemical observations.

| Sr. No. | Genotype | Hair Density (Trichomes/cm²) | | | | No. of Gossypol Glands/cm² | | | | Chlorophyll Content Index | | | | Total Soluble Sugar (%) | | | | Total Nitrogen (%) | | | | Crude Protein (%) | | | | No. of Sympo-dia/Plant | No. of Bolls per Plant | Boll Weight (g) | Seed Cotton Yield (g) |
|---|---|---|---|---|---|---|---|---|---|---|---|---|---|---|---|---|---|---|---|---|---|---|---|---|---|---|---|---|---|
| | | 45 DAS | 60 DAS | 90 DAS | 120 DAS | 45 DAS | 60 DAS | 90 DAS | 120 DAS | 45 DAS | 60 DAS | 90 DAS | 120 DAS | 45 DAS | 60 DAS | 90 DAS | 120 DAS | 45 DAS | 60 DAS | 90 DAS | 120 DAS | 45 DAS | 60 DAS | 90 DAS | 120 DAS | | | | |
| SGF_001 | Na-maskar_Gold-81 | 75.99 | 57.77 | 134.34 | 70.78 | 30.44 | 16.77 | 25.45 | 11.76 | 32.11 | 51.38 | 24.01 | 56.48 | 29.63 | 28.96 | 30.28 | 26.52 | 1.41 | 1.39 | 0.94 | 1.32 | 3.53 | 3.47 | 2.34 | 3.29 | 14.77 | 9.73 | 3.27 | 24.17 |
| SGF_002 | AV-G13 | 97.32 | 64.77 | 89.76 | 58.10 | 32.44 | 23.53 | 24.99 | 14.21 | 29.51 | 61.33 | 30.73 | 63.82 | 26.76 | 25.80 | 28.83 | 33.08 | 1.35 | 1.30 | 0.98 | 1.39 | 3.38 | 3.26 | 2.44 | 3.48 | 11.50 | 7.73 | 1.93 | 21.50 |
| SGF_003 | AV-G11 | 97.77 | 66.67 | 74.99 | 52.67 | 33.67 | 16.89 | 16.87 | 9.53 | 29.47 | 54.47 | 27.73 | 64.81 | 24.72 | 23.95 | 27.15 | 34.04 | 1.34 | 1.30 | 0.92 | 1.41 | 3.35 | 3.25 | 2.29 | 3.51 | 11.90 | 8.93 | 3.20 | 23.50 |
| SGF_006 | PA-255 | 88.00 | 71.89 | 91.56 | 125.11 | 11.78 | 50.11 | 24.56 | 18.43 | 31.02 | 63.61 | 28.63 | 37.88 | 26.77 | 20.81 | 29.89 | 27.06 | 1.37 | 1.35 | 0.91 | 1.12 | 3.42 | 3.36 | 2.28 | 2.80 | 12.40 | 8.20 | 2.43 | 19.83 |
| SGF_008 | AV-C14 | 130.22 | 63.78 | 107.09 | 64.00 | 57.67 | 21.89 | 23.21 | 12.76 | 24.87 | 35.02 | 22.97 | 30.85 | 18.70 | 25.80 | 26.38 | 29.76 | 1.38 | 1.31 | 0.70 | 0.93 | 3.46 | 3.28 | 1.74 | 2.32 | 12.63 | 9.00 | 2.50 | 21.67 |
| SGF_010 | GA-8004 | 88.56 | 57.67 | 121.77 | 88.22 | 14.44 | 46.44 | 20.89 | 16.77 | 37.52 | 41.07 | 24.10 | 55.86 | 30.65 | 28.88 | 29.02 | 20.58 | 1.48 | 1.38 | 0.85 | 1.40 | 3.70 | 3.46 | 2.13 | 3.51 | 15.10 | 9.73 | 3.37 | 24.67 |
| SGF_301 | Suraj | 73.22 | 54.10 | 61.76 | 72.32 | 51.78 | 45.11 | 31.43 | 19.11 | 40.76 | 41.35 | 24.47 | 43.72 | 36.64 | 28.85 | 26.14 | 27.59 | 1.55 | 1.39 | 0.56 | 1.30 | 3.87 | 3.48 | 1.40 | 3.26 | 12.17 | 8.80 | 3.55 | 23.83 |
| SGF_303 | Chetna_J1 | 75.77 | 54.56 | 63.67 | 62.88 | 42.65 | 43.77 | 17.66 | 24.42 | 48.20 | 42.67 | 31.41 | 47.45 | 37.28 | 30.48 | 38.82 | 25.57 | 1.57 | 1.41 | 1.00 | 1.32 | 3.92 | 3.52 | 2.50 | 3.29 | 12.20 | 7.33 | 2.43 | 19.50 |
| SGF_305 | Chetna D11 | 144.10 | 39.45 | 78.42 | 74.11 | 44.27 | 17.66 | 33.32 | 15.77 | 30.01 | 49.82 | 27.24 | 47.19 | 19.09 | 33.44 | 32.27 | 21.68 | 1.36 | 1.49 | 0.98 | 1.24 | 3.40 | 3.73 | 2.45 | 3.10 | 12.97 | 10.00 | 3.39 | 23.40 |
| SGF_306 | Chetna-S1 | 44.11 | 43.88 | 65.09 | 35.20 | 28.89 | 33.56 | 20.45 | 23.09 | 34.08 | 48.70 | 30.18 | 61.20 | 27.79 | 30.71 | 33.02 | 29.69 | 1.43 | 1.48 | 1.00 | 1.36 | 3.58 | 3.71 | 2.50 | 3.39 | 12.73 | 8.00 | 2.50 | 21.63 |
| SGF_318 | GH–8032 | 50.78 | 25.43 | 42.42 | 25.67 | 44.89 | 16.44 | 17.53 | 20.33 | 32.17 | 50.70 | 26.48 | 60.87 | 27.29 | 34.97 | 38.63 | 30.67 | 1.42 | 1.52 | 1.13 | 1.37 | 3.54 | 3.79 | 2.82 | 3.42 | 12.93 | 6.40 | 3.55 | 22.17 |
| SGF_319 | GH-8001 E | 122.11 | 79.32 | 63.63 | 84.44 | 44.22 | 27.11 | 32.20 | 23.09 | 28.33 | 37.89 | 26.12 | 37.35 | 18.89 | 25.61 | 30.04 | 26.27 | 1.38 | 1.39 | 0.98 | 0.97 | 3.45 | 3.47 | 2.45 | 2.41 | 13.53 | 8.93 | 3.37 | 24.17 |
| SGF_321 | BKMP_27 | 130.34 | 78.21 | 67.87 | 60.89 | 44.89 | 41.33 | 30.42 | 14.22 | 21.25 | 59.62 | 25.88 | 32.21 | 18.93 | 26.94 | 30.64 | 21.38 | 1.35 | 1.33 | 0.99 | 0.87 | 3.38 | 3.33 | 2.47 | 2.17 | 12.80 | 7.60 | 2.07 | 20.83 |
| SGF_322 | AKH 09-5 | 131.33 | 61.67 | 68.34 | 48.00 | 54.89 | 31.33 | 20.52 | 14.76 | 20.51 | 47.65 | 25.15 | 36.18 | 18.78 | 25.19 | 29.73 | 20.74 | 1.37 | 1.30 | 0.95 | 0.60 | 3.44 | 3.26 | 2.37 | 1.50 | 16.10 | 10.90 | 4.00 | 24.80 |
| SGF_323 | AKH 9916 | 135.44 | 90.11 | 131.67 | 99.22 | 57.44 | 36.11 | 23.55 | 17.67 | 20.40 | 34.38 | 23.29 | 39.55 | 19.00 | 22.73 | 26.50 | 34.33 | 1.37 | 1.28 | 0.59 | 0.93 | 3.43 | 3.19 | 1.47 | 2.32 | 13.83 | 7.40 | 2.28 | 19.83 |
| SGF_701 | Mallika | 82.22 | 34.09 | 61.45 | 31.09 | 36.67 | 47.00 | 23.55 | 20.20 | 39.06 | 40.57 | 28.96 | 37.46 | 25.67 | 28.71 | 26.67 | 25.72 | 1.46 | 1.40 | 0.68 | 0.97 | 3.65 | 3.49 | 1.71 | 2.41 | 11.43 | 9.80 | 3.90 | 22.00 |
| SGF_705 | Nirmal 996 | 100.88 | 45.43 | 42.43 | 61.66 | 50.56 | 49.43 | 33.34 | 21.53 | 32.49 | 39.67 | 27.05 | 39.01 | 26.98 | 27.64 | 26.85 | 20.22 | 1.38 | 1.37 | 0.69 | 0.97 | 3.45 | 3.43 | 1.73 | 2.41 | 16.17 | 10.87 | 4.07 | 28.33 |
| SGF_723 | Vasudha-1318 | 83.66 | 36.00 | 54.66 | 29.32 | 54.00 | 47.22 | 31.55 | 14.21 | 40.10 | 39.83 | 23.56 | 56.20 | 35.85 | 29.91 | 28.94 | 28.83 | 1.52 | 1.41 | 0.94 | 1.32 | 3.81 | 3.51 | 2.35 | 3.30 | 9.77 | 6.93 | 1.97 | 15.00 |
| SGF_718 | Gold Bhakti 245 | 59.77 | 35.89 | 94.89 | 28.54 | 12.67 | 37.76 | 34.32 | 20.77 | 55.89 | 38.48 | 25.34 | 60.13 | 39.67 | 30.93 | 29.57 | 31.73 | 1.63 | 1.36 | 0.90 | 1.39 | 4.07 | 3.41 | 2.24 | 3.47 | 11.33 | 6.80 | 3.93 | 20.17 |
| SGF_719 | Raja 954 | 85.10 | 41.87 | 64.43 | 19.54 | 45.22 | 41.56 | 37.87 | 19.99 | 37.67 | 36.47 | 27.73 | 64.71 | 31.66 | 24.71 | 31.58 | 33.68 | 1.49 | 1.35 | 0.93 | 1.38 | 3.73 | 3.38 | 2.33 | 3.45 | 9.53 | 6.53 | 1.90 | 12.33 |
| Check | DCH–32 | 48.43 | 21.99 | 37.87 | 44.00 | 11.78 | 15.78 | 16.87 | 21.20 | 63.78 | 63.61 | 31.47 | 45.01 | 42.89 | 37.06 | 39.03 | 19.13 | 1.71 | 1.55 | 1.15 | 1.30 | 4.29 | 3.88 | 2.87 | 3.26 | 10.30 | 5.87 | 2.33 | 18.00 |
| Check | DHY–286 | 144.00 | 101.10 | 139.88 | 135.56 | 38.33 | 41.87 | 30.99 | 18.63 | 47.18 | 32.71 | 24.02 | 31.60 | 29.89 | 21.26 | 26.96 | 23.97 | 1.38 | 1.26 | 0.76 | 0.96 | 3.46 | 3.15 | 1.91 | 2.40 | 12.20 | 6.40 | 3.97 | 23.17 |
| | Range | 44.11– 144.10 | 21.99– 101.1 | 37.87– 139.88 | 19.54– 135.56 | 11.78– 57.67 | 15.78– 50.11 | 16.87– 37.87 | 9.53– 24.42 | 20.40– 63.78 | 32.71– 63.61 | 22.97– 31.47 | 30.85– 64.81 | 18.7– 42.89 | 20.81– 37.06 | 26.14– 39.03 | 19.13– 34.33 | 1.34– 1.71 | 1.26– 1.55 | 0.56– 1.15 | 0.6– 1.41 | 3.35– 4.29 | 3.15– 3.88 | 1.4– 2.87 | 1.5– 3.51 | 9.53– 16.17 | 5.87– 10.9 | 1.9– 4.07 | 12.33– 28.33 |
| | Mean | 94.96 | 55.71 | 79.91 | 62.33 | 38.34 | 34.03 | 25.98 | 17.84 | 35.29 | 45.95 | 26.66 | 47.71 | 27.89 | 27.88 | 30.32 | 26.92 | 1.44 | 1.38 | 0.89 | 1.17 | 3.60 | 3.45 | 2.22 | 2.93 | 12.65 | 8.27 | 3.00 | 21.57 |
| | S.E. (M)± | 0.29 | 0.50 | 0.74 | 1.85 | 0.49 | 0.41 | 0.43 | 0.40 | 0.84 | 1.23 | 1.25 | 1.46 | 0.20 | 0.28 | 0.24 | 0.19 | 0.01 | 0.01 | 0.02 | 0.02 | 0.03 | 0.02 | 0.04 | 0.04 | 0.50 | 0.40 | 0.18 | 1.00 |
| | C.D. 5% | 0.84 | 1.43 | 2.10 | 5.29 | 1.41 | 1.17 | 1.22 | 1.14 | 2.40 | 3.50 | 3.55 | 4.16 | 0.56 | 0.79 | 0.68 | 0.54 | 0.04 | 0.02 | 0.05 | 0.04 | 0.09 | 0.06 | 0.12 | 0.10 | 1.42 | 1.15 | 0.52 | 2.86 |
| | C.V. | 0.53 | 1.56 | 1.60 | 5.15 | 2.23 | 2.09 | 2.84 | 3.89 | 4.13 | 4.63 | 8.09 | 5.30 | 1.22 | 1.73 | 1.37 | 1.21 | 1.49 | 1.01 | 3.16 | 2.16 | 1.49 | 1.01 | 3.16 | 2.16 | 6.79 | 8.40 | 10.53 | 8.05 |

2.3.2. Total Nitrogen (Kjeldhal Method)

Firstly, for digestion, the leaf sample was transferred to the digestion tube and the catalyst mixture and concentrated $H_2SO_4$ was added to the tubes which were kept in the block digester and heated at 100 °C until the digestion was completed and a transparent liquid was obtained. The tube was then removed from the block digester and was cooled by adding a sufficient quantity of distilled water. Secondly, for the distillation process, boric acid solution was taken in a conical flask and was placed in such a way that the condenser outlet of the distillation apparatus was dipped in the boric acid solution. Then, 10 mL of digested aliquot was taken in the distillation tube of the Kjeldhal distillation apparatus, and NaOH was added with the help of the apparatus. The distillation was then carried out for 9 min and the color of the boric acid in the conical flask changed from pink to green. It was then titrated drop-by-drop with 0.1 N $H_2SO_4$ until the boric acid changed the color to pink. The reading on the burette was recorded and calculations were performed according to the given formula:

$$\text{Total Nitrogen} = (A - B) \times 0.014 \times \text{Normality of acid} \times \frac{100}{\text{Weight of sample}} \quad (2)$$

where A = Reading of plant sample solution; B = Reading of blank sample solution.

2.3.3. Crude Protein Content of Leaf

The protein content was estimated by using the nitrogen to protein conversion factor, i.e., 6.25:

$$\text{Crude Protein} = 6.25 \times \text{Total Nitrogen}$$

*2.4. Morphological Observations*

Morphological observations, such as the number of bolls per plant, boll weight, number of sympodia (reproductive branches) and seed cotton yield, were recorded by visual observations at maturity (Table 2).

**3. Results**

The 2019–2020 study was conducted to observe the sucking pest incidence on the cotton genotypes which was found to be effective, hence, a more comprehensive and detailed study was carried out in the following year, i.e., 2020–2021.

*3.1. Sucking Pest Infestation*

The sucking pest population was studied on the 22 cotton genotypes in two years, 2019–2020 and 2020–2021, at the same location. The pooled analysis of the data was performed and it was inferred that a significant genotype x environment interaction for all the genotypes under study existed (Table 3). Aphid population was observed to be the lowest in the genotype AV-G11 (4.92 aphids/leaf) and Namaskar Gold_81 (5.21 aphids/leaf) at 30 DAS, the genotype AV-G11 (1.78 aphids/leaf) and PA-255 (1.74 aphids/leaf) at 60 DAS, and the genotype PA-255 (0.74 aphids/leaf) and GA-8004 (0.93 aphids/leaf) at 90 DAS. The *Cicadellidae* population was recorded at a minimum on the genotypes Nirmal 996 (0.47 aphids/leaf) and AKH-9916 (0.49 *Cicadellidae*/leaf) at 30 DAS, the genotype AKH-09-5 (0.66 *Cicadellidae*/leaf) and AV-C14 (0.78 *Cicadellidae*/leaf) at 60 DAS, and genotype PA-255 (0.59 *Cicadellidae*/leaf) and AV-G13 (0.77 *Cicadellidae*/leaf) at 90 DAS. The thrips' minimum incidence was recorded on the genotype AV-C14 (0.47 thrips/leaf) and AKH-0905 (0.52 thrips/leaf) at 30 DAS, the genotype PA-255 (1.09 thrips/leaf) and AV-G13 (1.8 thrips/leaf) at 60 DAS, and the genotype GA-8004 (0.75 thrips/leaf) and AV-G13 (0.96 thrips/leaf) at 90 DAS. The whitefly studies showed that a minimum population was observed at 30 DAS on the check DCH-32 (0.21 whitefly/leaf) and genotype Chetna_S1 (0.28 whitefly/leaf), at 60 DAS on the genotype PA-255 (1.01 whitefly/leaf) and AV-G13 (1.27 whitefly/leaf), and at 90 DAS on the genotype AV-G11 (0.78 whitefly/leaf) and AV-G13 (0.82 whitefly/leaf). The study of the averages (Table 3), inferred that the aphid

population was initially higher at 30 DAS and decreased towards 90 DAS; for *Cicadellidae*, the population was observed to increase from 30 DAS to 90 DAS, whereas for thrips and whitefly, the highest mean population was recorded at 60 DAS.

The study found that *G. arboreum* genotypes, specifically hybrid Namaskar Gold_81 and other varieties, such as AV-G13, AV-G11, PA-255, AV-C14, and GA-8004, exhibited lower populations of sucking pests compared to *G. hirsutum* varieties and hybrids. This observation is supported by the graphs (Figures 6–9), which visually depict the lower incidence of sucking pests in *G. arboreum* genotypes, compared to *G. hirsutum* genotypes (Table S1). These findings provide evidence that *G. arboreum* genotypes have a natural advantage in terms of reduced susceptibility to sucking pests, reinforcing their potential suitability for cotton cultivation.

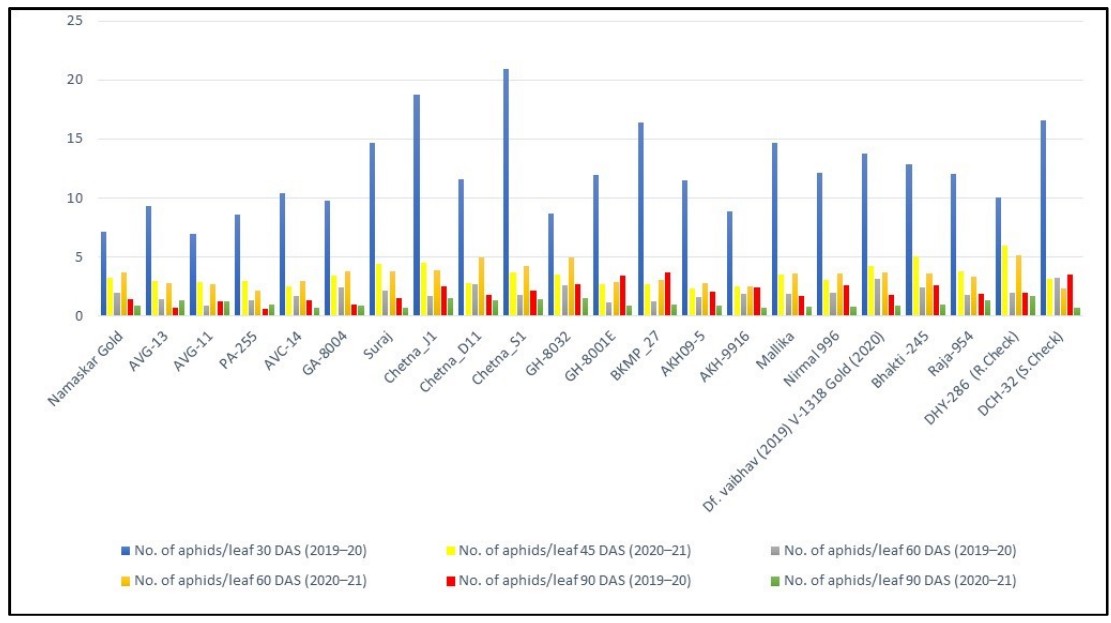

**Figure 6.** Number of Aphids per leaf.

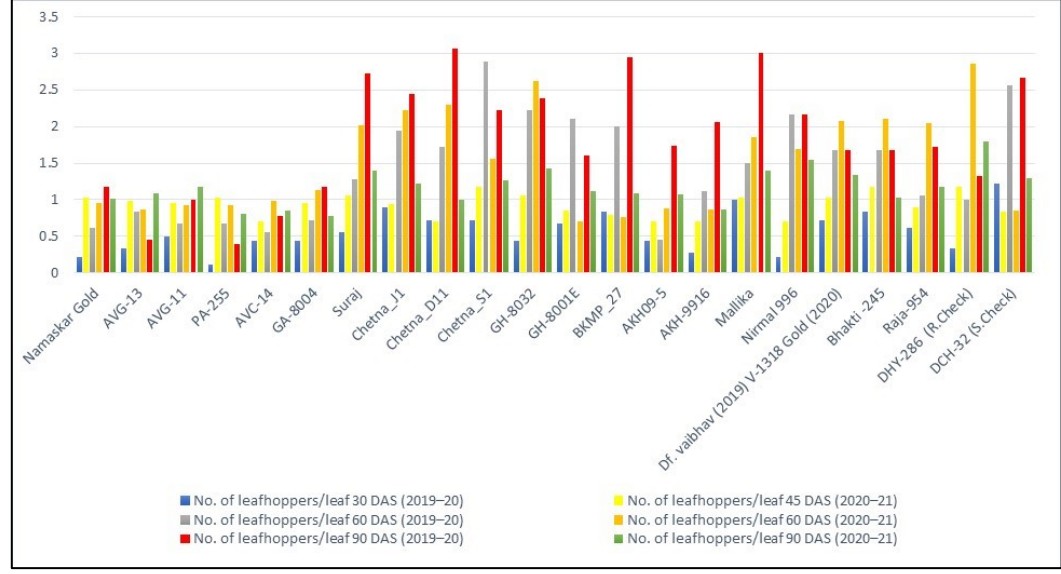

**Figure 7.** Number of Cicadellidae (leafhoppers) per leaf.

**Table 3.** Pooled sucking pest data of the two years 2019–2020 and 2020–2021.

| Sr. No. | Cultivar Code | Culitvar Name | No. of Aphid/Leaf | | | No. of *Cicadellidae*/Leaf | | | No. of Thrips/Leaf | | | No. of Whitefly/Leaf | | |
|---|---|---|---|---|---|---|---|---|---|---|---|---|---|---|
| | | Genotypes | 30 DAS | 60 DAS | 90 DAS | 30 DAS | 60 DAS | 90 DAS | 30 DAS | 60 DAS | 90 DAS | 30 DAS | 60 DAS | 90 DAS |
| 1 | SGF_001 | Namaskar Gold | 5.21 * | 2.87 | 1.16 | 0.62 | 0.79 | 1.09 | 0.91 | 3.65 | 1.37 | 0.57 | 1.30 | 1.65 |
| 2 | SGF_002 | AV-G13 | 6.12 | 2.11 | 1.02 | 0.66 | 0.85 | 0.77 * | 0.88 | 1.80 * | 0.96 * | 0.76 | 1.27 * | 0.82 * |
| 3 | SGF_003 | AV-G11 | 4.92 * | 1.78 * | 1.25 | 0.73 | 0.80 | 1.08 | 0.81 | 1.88 | 1.02 | 0.79 | 1.82 | 0.78 * |
| 4 | SGF_006 | PA-255 | 5.80 | 1.74 * | 0.74 * | 0.57 | 0.79 | 0.59 * | 0.78 | 1.09 * | 1.24 | 0.63 | 1.10 * | 1.12 |
| 5 | SGF_008 | AV-C14 | 6.45 | 2.35 | 1.01 | 0.58 | 0.78 * | 0.82 | 0.47 * | 3.20 | 1.00 | 0.89 | 1.80 | 1.12 |
| 6 | SGF_010 | GA-8004 | 6.61 | 3.13 | 0.93 * | 0.70 | 0.92 | 0.97 | 1.18 | 3.23 | 0.75 * | 0.51 | 1.93 | 1.10 |
| 7 | SGF_301 | Suraj | 9.54 | 2.93 | 1.08 | 0.81 | 1.65 | 2.05 | 1.32 | 3.79 | 2.02 | 0.65 | 1.80 | 1.32 |
| 8 | SGF_303 | Chetna_J1 | 11.62 | 2.81 | 2.01 | 0.92 | 2.09 | 1.83 | 1.41 | 3.73 | 2.70 | 0.62 | 2.03 | 1.87 |
| 9 | SGF_305 | Chetna D11 | 7.18 | 3.81 | 1.56 | 0.72 | 2.01 | 2.03 | 0.64 | 3.71 | 2.75 | 0.79 | 1.52 | 1.68 |
| 10 | SGF_306 | Chetna-S1 | 12.31 | 3.02 | 1.79 | 0.95 | 2.23 | 1.75 | 0.90 | 3.71 | 1.71 | 0.28 * | 1.85 | 2.14 |
| 11 | SGF_318 | GH-8032 | 6.10 | 3.81 | 2.10 | 0.75 | 2.42 | 1.91 | 1.11 | 3.66 | 1.54 | 0.45 | 1.41 | 2.30 |
| 12 | SGF_319 | GH-8001 E | 7.33 | 2.00 | 2.15 | 0.76 | 1.41 | 1.37 | 0.67 | 3.03 | 2.05 | 0.59 | 2.24 | 1.30 |
| 13 | SGF_321 | BKMP_27 | 9.53 | 2.17 | 2.33 | 0.81 | 1.38 | 2.02 | 0.64 | 3.29 | 2.51 | 0.78 | 2.36 | 2.14 |
| 14 | SGF_322 | AKH 09-5 | 6.92 | 2.21 | 1.48 | 0.58 | 0.66 * | 1.40 | 0.52 * | 2.04 | 1.83 | 0.81 | 2.13 | 1.28 |
| 15 | SGF_323 | AKH 9916 | 5.67 | 2.18 | 1.53 | 0.49 * | 0.99 | 1.46 | 0.62 | 1.83 | 1.86 | 0.92 | 2.42 | 2.50 |
| 16 | SGF_701 | Mallika | 9.10 | 2.73 | 1.23 | 1.02 | 1.68 | 2.20 | 1.09 | 3.60 | 1.47 | 0.54 | 1.79 | 2.42 |
| 17 | SGF_705 | Nirmal 996 | 7.59 | 2.78 | 1.67 | 0.47 * | 1.93 | 1.86 | 0.82 | 3.69 | 2.05 | 0.59 | 2.06 | 2.06 |
| 18 | SGF_723 | Vasudha-1318 Gold | 9.03 | 3.44 | 1.32 | 0.88 | 1.87 | 1.50 | 1.30 | 3.90 | 1.66 | 0.63 | 2.15 | 1.90 |
| 19 | SGF_718 | Bhakti 245 | 8.92 | 2.98 | 1.77 | 1.01 | 1.88 | 1.35 | 1.57 | 3.68 | 1.55 | 0.66 | 1.62 | 1.72 |
| 20 | SGF_719 | Raja 954 | 7.91 | 2.57 | 1.56 | 0.75 | 1.56 | 1.45 | 1.24 | 3.77 | 1.60 | 0.32 | 1.46 | 1.73 |
| 21 | Check | DCH-32 | 8.00 | 3.55 | 1.83 | 0.75 | 1.93 | 1.57 | 1.45 | 3.56 | 1.48 | 0.21 * | 1.99 | 1.26 |
| 22 | Check | DHY-286 | 9.85 | 2.77 | 2.10 | 1.03 | 1.70 | 1.99 | 1.10 | 2.09 | 3.12 | 0.77 | 2.12 | 2.73 |
| | | Mean | 7.80 | 2.71 | 1.53 | 0.75 | 1.47 | 1.50 | 0.97 | 3.09 | 1.74 | 0.63 | 1.83 | 1.68 |
| | | Range | 4.92–12.31 | 1.74–3.81 | 0.74–2.33 | 0.47–1.03 | 0.66–2.42 | 0.59–2.20 | 0.47–1.57 | 1.09–3.90 | 0.75–3.12 | 0.21–0.92 | 1.10–2.42 | 0.78–2.73 |
| | | SE (M)± | 0.61 | 0.23 | 0.21 | 0.07 | 0.19 | 0.17 | 0.09 | 0.20 | 0.23 | 0.05 | 0.18 | 0.19 |
| | | CD at 5% | 1.75 | 0.65 | 0.59 | 0.20 | 0.54 | 0.48 | 0.26 | 0.57 | 0.66 | 0.14 | 0.50 | 0.56 |
| | | CV | 10.25 | 10.36 | 10.14 | 8.23 | 10.68 | 10.71 | 5.87 | 10.72 | 9.68 | 7.88 | 7.84 | 8.78 |

Note: DAS: Days after sowing; SGF—Seeding the green future; *—lowest sucking pest population.

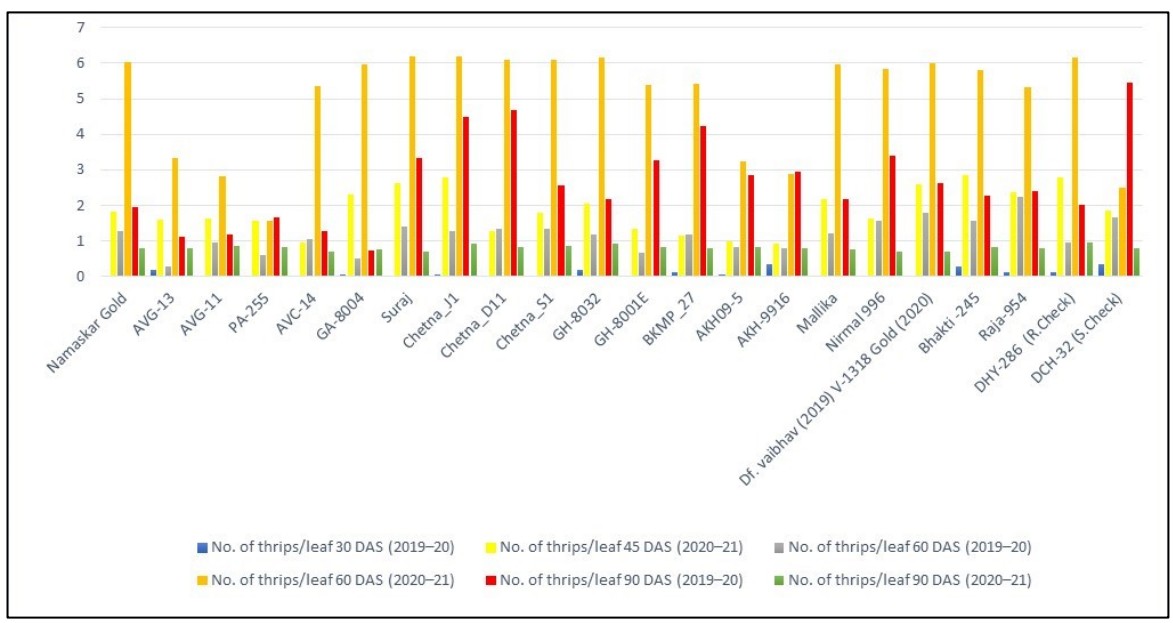

**Figure 8.** Number of thrips per leaf.

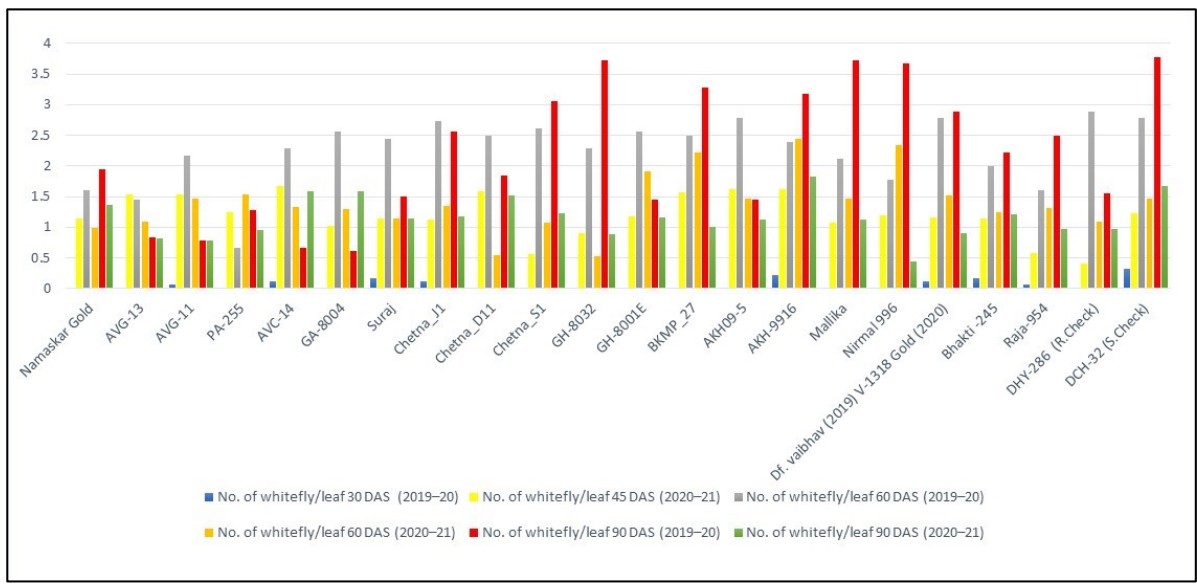

**Figure 9.** Number of whitefly per leaf.

*3.2. Correlation Studies*

The genotypic and phenotypic correlations were calculated based on the data of the sucking pest and other relevant characters, such as gossypol glands (Figure 4), trichome density (Figure 5), etc. The positive value of r (correlation coefficient) indicated change of the two variables in the same direction, whereas the variables' negative value of r changed in the opposite direction. The degree and direction of the association between the characters was studied [11].

3.2.1. Correlations with the Number of Aphids per Leaf

The correlation studies conducted in the research revealed significant findings regarding the relationship between aphids and various factors (Table 4). Firstly, there was a negative and significant correlation between aphids and gossypol glands at 45, 60, and 90 days after sowing (DAS), indicating that higher gossypol gland density in cotton plants is associated with reduced aphid populations. Trichome density also exhibited a negative

and significant correlation with the aphid population at 45, 60, and 90 DAS, suggesting that higher trichome density serves as a physical defense mechanism against aphid infestation. On the other hand, chlorophyll content showed a significant positive correlation with aphids at 45 DAS (0.9371 and 0.8868 at genotypic and phenotypic levels, respectively) and 90 DAS (0.8515 and 0.6396 at genotypic and phenotypic levels, respectively), as well as with biochemical characters such as total soluble sugar, total nitrogen, and crude protein. Regarding morphological characters, aphids displayed negative and significant correlations with the number of bolls per plant at 45 DAS (−0.5656 and −0.4641 at genotypic and phenotypic levels, respectively) and 90 DAS (−0.4637 and −0.3869 at genotypic and phenotypic levels, respectively), as well as with seed cotton yield at 45 DAS (−0.444 and −0.3701 at genotypic and phenotypic levels, respectively) and 90 DAS (−0.3346 and −0.2693 at genotypic and phenotypic levels, respectively). These correlations provide valuable insights into the factors influencing aphid populations and their impact on cotton plants' physiology, morphology, and productivity.

### 3.2.2. Correlations with the Number of Cicadellidae per Leaf

*Cicadellidae* (leafhoppers) showed negative and significant correlation (Table 4), with gossypol glands on 45 DAS 45 (−0.7111 and −0.5776 at genotypic and phenotypic level, respectively) but had a negative and significant correlation overall with trichome density. However, chlorophyll content showed a positive and significant correlation at 45 and 90 DAS with *Cicadellidae*. Total soluble sugar, total nitrogen and crude protein had a positive and significant correlation at 45 and 60 DAS with *Cicadellidae*. *Cicadellidae* at 45 DAS (−0.4806 and −0.3823 at genotypic and phenotypic level, respectively) and 60 DAS (−0.4345 and −0.3681 at genotypic and phenotypic level, respectively) had a negative and significant correlation with the number of bolls per plant.

### 3.2.3. Correlations with the Number of Thrips per Leaf

The correlation study of thrips recorded a negative and significant correlation with gossypol glands at 90 DAS (−0.6474 and −0.4806 at genotypic and phenotypic level, respectively), and trichome density had a negative and significant correlation at 45 and 60 DAS with thrips (Table 4). Chlorophyll content had a positive and significant correlation at 45 and 90 DAS with thrips. Biochemical correlation recorded that total soluble sugar had a positive and significant correlation at 45, 60 and 90 DAS with thrips. However, total nitrogen and crude protein had an overall positive and significant correlation with thrips. In relation to the morphological characters studied, the number of bolls per plant had a negative and significant correlation at 45 DAS (−0.4912 and −0.4219 at genotypic and phenotypic level, respectively) and 90 DAS (−0.5199 and −0.3128 at genotypic and phenotypic level, respectively). Boll weight and seed cotton yield had a negative and non-significant correlation with thrips.

### 3.2.4. Correlations with the Number of Whiteflies per Leaf

Whiteflies had a positive and significant correlation (Table 4), with gossypol glands at 45 DAS (0.4557 and 0.4356 at genotypic and phenotypic level, respectively) and 60 DAS (0.4266 and 0.412 at genotypic and phenotypic level, respectively). Trichome density, however, had a positive and significant correlation with whiteflies overall. The chlorophyll content had a negative and significant correlation at 45 DAS (−0.6784 and −0.648 at genotypic and phenotypic level, respectively) and 90 DAS (−0.6121 and −0.3916 at genotypic and phenotypic level, respectively) with whiteflies, which indicated that less chlorophyll content had more whitefly incidence. Total soluble sugar had a negative and significant correlation at 45 and 60 DAS with whiteflies, whereas total nitrogen and crude protein, on the other hand, had a negative and significant correlation overall.

**Table 4.** Correlation of sucking pest with morpho-physiological, biochemical and morphological characters at 45, 60, 90 and 120 DAS.

| Character | | Correlation | Trichome Density/cm² | Gossypol Glands/cm² | Chlorophyll Content Index | Total Soluble Sugar (%) | Total Nitrogen (%) | Crude Protein (%) | No. of Bolls per Plant 120 DAS | Boll Weight 120 DAS | Seed Cotton Yield 120 DAS |
|---|---|---|---|---|---|---|---|---|---|---|---|
| No. of aphids/leaf | 45 DAS | G | −0.7164 ** | −0.4755 ** | 0.9371 ** | 0.9537 ** | 0.991 ** | 0.991 ** | −0.5656 ** | −0.1132 | −0.444 * |
| | | P | −0.6684 ** | −0.4383 ** | 0.8562 ** | 0.8868 ** | 0.9342 ** | 0.9342 ** | −0.4641 ** | −0.0867 | −0.3701 ** |
| | 60 DAS | G | −0.8233 ** | −0.3853 | 0.2383 | 0.9984 ** | 0.9832 ** | 0.9832 ** | −0.1586 | 0.0763 | −0.0353 |
| | | P | −0.742 ** | −0.3465 ** | 0.2042 | 0.8934 ** | 0.8803 ** | 0.8803 ** | −0.1277 | 0.0203 | −0.0954 |
| | 90 DAS | G | −0.4731 * | −0.4524 * | 0.8515 ** | 0.8654 ** | 0.8201 ** | 0.8201 ** | −0.4637 * | −0.3549 | −0.3346 |
| | | P | −0.4542 ** | −0.4299 ** | 0.6396 ** | 0.8364 ** | 0.7806 ** | 0.7806 ** | −0.3869 ** | −0.3189 ** | −0.2693 * |
| | 120 DAS | G | −0.5632 ** | −0.0695 | 0.8758 ** | 0.6692 ** | 0.7431 ** | 0.7431 ** | −0.477 * | −0.2486 | −0.4406 * |
| | | P | −0.5401 ** | −0.0751 | 0.8327 ** | 0.6525 ** | 0.7186 ** | 0.7186 ** | −0.4133 ** | −0.2255 | −0.3663 ** |
| No. of *cicadellidae*/leaf (*cicadellidae*) | 45 DAS | G | −0.9662 ** | −0.7111 ** | −0.6891 ** | 0.77 ** | 0.6791 ** | 0.6791 ** | −0.4806 * | −0.0788 | −0.3186 |
| | | P | −0.8036 ** | −0.5776 ** | −0.5799 ** | 0.6356 ** | 0.564 ** | 0.564 ** | −0.3823 ** | −0.0753 | −0.2418 |
| | 60 DAS | G | −0.8328 ** | −0.1454 | 0.0899 | 0.8482 ** | 0.8144 ** | 0.8144 ** | −0.4345 * | −0.004 | −0.3888 |
| | | P | −0.813 ** | −0.1431 | 0.0881 | 0.8259 ** | 0.7845 ** | 0.7845 ** | −0.3681 ** | −0.0261 | −0.3073 * |
| | 90 DAS | G | −0.6887 ** | −0.1105 | 0.4711 ** | 0.4214 | 0.2634 | 0.2634 | −0.3134 | 0.104 | −0.0681 |
| | | P | −0.5907 ** | −0.0828 | 0.3307 ** | 0.3523 ** | 0.2016 | 0.2016 | −0.2407 | 0.1422 | −0.0322 |
| | 120 DAS | G | −0.9148 ** | 0.3464 | 0.6249 ** | 0.336 | 0.5375 ** | 0.5375 ** | −0.5054 * | −0.111 | −0.6575 ** |
| | | P | −0.7309 ** | 0.2691 * | 0.4883 ** | 0.2857 * | 0.4372 ** | 0.4372 ** | −0.4392 ** | −0.4972 ** | −0.4811 ** |
| No. of thrips/leaf | 45 DAS | G | −0.725 ** | −0.3927 | 0.898 ** | 0.9718 ** | 0.924 ** | 0.924 ** | −0.4912 * | −0.0076 | −0.4762 * |
| | | P | −0.6835 ** | −0.3657 ** | 0.8454 ** | 0.913 ** | 0.871 ** | 0.871 ** | −0.4219 ** | −0.0209 | −0.318 ** |
| | 60 DAS | G | −0.7181 ** | −0.024 | −0.1387 | 0.8076 ** | 0.7464 ** | 0.7464 ** | 0.0209 | 0.0912 | −0.0381 |
| | | P | −0.6897 ** | −0.0238 | −0.1294 | 0.7734 ** | 0.7059 ** | 0.7059 ** | 0.0107 | 0.046 | −0.0357 |
| | 90 DAS | G | −0.3423 | −0.6474 ** | 0.8687 ** | 0.9589 ** | 0.9237 ** | 0.9237 ** | −0.5199 * | −0.1594 | −0.1958 |
| | | P | −0.2464 * | −0.4806 ** | 0.4551 ** | 0.6732 ** | 0.6294 ** | 0.6294 ** | −0.3128 * | −0.1095 | −0.1469 |
| | 120 DAS | G | −0.0749 | −0.1698 | 0.3744 | 0.1513 | 0.491 * | 0.491 * | 0.0805 | −0.1409 | −0.0713 |
| | | P | −0.0652 | −0.1482 | 0.3298 ** | 0.131 | 0.4152 ** | 0.4152 ** | −0.1766 | −0.194 | −0.1333 |

**Table 4.** *Cont.*

| Character | | Correlation | Trichome Density/cm² | Gossypol Glands/cm² | Chlorophyll Content Index | Total Soluble Sugar (%) | Total Nitrogen (%) | Crude Protein (%) | No. of Bolls per Plant 120 DAS | Boll Weight 120 DAS | Seed Cotton Yield 120 DAS |
|---|---|---|---|---|---|---|---|---|---|---|---|
| No. of white-fly/leaf | 45 DAS | G | 0.7912 ** | 0.4557 * | −0.6784 ** | −0.6799 ** | −0.6287 ** | −0.6287 ** | 0.4195 | 0.0072 | 0.3232 |
| | | P | 0.7638 ** | 0.4356 ** | −0.648 ** | −0.6551 ** | −0.6085 ** | −0.6085 ** | 0.3598 ** | 0.024 | 0.2907 * |
| | 60 DAS | G | 0.5133 * | 0.4266 * | −0.2264 | −0.498 * | −0.4917 * | −0.4917 * | 0.1465 | −0.087 | 0.1366 |
| | | P | 0.4977 ** | 0.412 ** | −0.2239 | −0.4824 ** | −0.4689 ** | −0.4689 ** | 0.1177 | −0.0664 | 0.1243 |
| | 90 DAS | G | 0.789 ** | −0.0065 | −0.6121 ** | −0.244 | −0.4108 | −0.4108 | 0.0352 | 0.0982 | 0.1084 |
| | | P | 0.7533 ** | −0.0042 | −0.3916 ** | −0.2368 | −0.3972 ** | −0.3972 ** | −0.0037 | 0.0765 | 0.0718 |
| | 120 DAS | G | 0.5621 ** | 0.0186 | −0.642 ** | −0.2436 | −0.7124 ** | −0.7124 ** | 0.3765 | 0.0893 | 0.4018 |
| | | P | 0.5512 ** | 0.0234 | −0.6161 ** | −0.2405 | −0.6975 ** | −0.6975 ** | 0.4057 ** | 0.4528 ** | 0.4518 ** |
| Seed cotton yield | 45 DAS | G | 0.2703 | 0.055 | −0.3191 | −0.4057 | −0.459 * | −0.459 * | 0.7796 ** | 0.7913 ** | 1 |
| | | P | 0.241 | 0.051 | −0.2836 * | −0.3632 ** | −0.388 ** | −0.388 ** | 0.6246 ** | 0.6117 ** | 1 |
| | 60 DAS | G | 0.2531 | −0.1303 | −0.0137 | −0.0687 | −0.1093 | −0.1093 | 0.7796 ** | 0.7913 ** | 1 |
| | | P | 0.2208 | −0.1203 | −0.0038 | −0.0586 | −0.0925 | −0.0925 | 0.6246 ** | 0.6117 ** | 1 |
| | 90 DAS | G | 0.1534 | −0.1423 | −0.1975 | −0.3176 | −0.2865 | −0.2865 | 0.7796 ** | 0.7913 ** | 1 |
| | | P | 0.1349 | −0.1291 | −0.1671 | −2706 * | −0.2464 * | −0.2464 * | 0.6246 ** | 0.6117 ** | 1 |
| | 120 DAS | G | 0.3616 | −0.0942 | −0.3176 | −0.4352 * | −0.3296 | −0.3296 | −0.7796 ** | 0.7913 ** | 1 |
| | | P | 0.3157 ** | −0.0956 | −0.2586 * | −0.3816 ** | −0.2957 * | −0.2957 * | 0.6246 ** | 0.6117 ** | 1 |

Note: **—Significant at 1% level of significance *—Significant at 5% level of significance.

## 4. Discussion

A significant genotype and environmental interaction were observed in the study of cotton genotypes conducted in two years: 2019-2020 and 2020-2021 at the same location. The lowest population of aphids was recorded at 90 DAS, whereas *Cicadellidae*, thrips and whiteflies were recorded at 30 DAS. The study highlighted certain tolerant genotypes (AV-G11, PA-255, GA-8004, AKH 09-5, AV-C14, and AV-G13) that exhibited the ability to withstand sucking pest infestation, suggesting their potential for pest management strategies for organic and sustainable cotton cultivation [12].

In the comparative study of the *arboreum* and hirsutum cotton genotypes, the *arboreum* genotypes (F1 hybrid Namaskar Gold_81 and cultivar PA-255 and newly developed varietal lines, like AV-G13, AV-G11, AV-C14, and GA-8004), recorded less sucking pest population in comparison to the *hirsutum* F1 hybrids and varieties. It was thus seen that *G. arboreum* genotypes reported the least sucking pest population, and the difference is visualized from the graphs (Figures 6–9). These findings emphasized the potential of *G. arboreum* as a promising solution for managing sucking pest infestation [13]. The natural tolerance displayed the inherent resistance mechanism of the genotypes, which can be further investigated to develop pest-resistant cotton varieties. Incorporation of *arboreum* genotypes into breeding programs and cultivation practices will definitely contribute to sustainable pest management and reduce the reliance on synthetic pesticides [14].

Correlation studies indicated a significantly positive correlation between whitefly population and gossypol glands at 45 and 60 DAS [15] and with overall trichome density [15–17]. The high density of trichomes were associated with increased leaf pubescence, facilitated enhanced whitefly oviposition and nymph development. Additionally, trichome density was found to elevate humidity at the leaf surface [18], creating favorable conditions for nymph development [19,20]. Reduced chlorophyll levels were observed as whitefly populations increased at 60 DAS, likely due to the insect's sap-sucking activity, particularly during the demanding reproductive growth phase [21,22]. Chlorophyll is a crucial pigment responsible for photosynthesis; whitefly feeding led to significant decrease in sugar levels, indicating a disruption in sugar metabolism. These changes in sugar content were associated with a decline in chlorophyll concentration, highlighting the interplay between sugar metabolism, chlorophyll synthesis, and whitefly infestation [23]. The population of pests, along with the timing of infestation, are the driving factors that may cause alterations in the chlorophyll content highlights the need for timely pest control interventions to ensure that the yield is not penalized at later stages. In general, for all the other sucking pests studied, including aphids, Cicadellidae, and thrips, a negatively significant correlation was observed with gossypol glands and trichome density. [24–26]. Furthermore, Gossypol gland per $cm^2$ and Trichome density per $cm^2$ emerged as a crucial factor influencing sucking pest tolerance, as it exhibited a significant correlation with insect pest populations. Higher trichome density and high-gossypol glands acted as a barrier against the sucking pest complex in cotton, excluding whiteflies. However, these findings were contrary to whitefly infestations, owing to oviposition habits.

Resistance to this sucking pest complex is possibly controlled by the inherent capacity of the plant to produce higher quantities of gossypol glands with enhanced trichome density. It is quite clear that parameters such as chlorophyll content, gossypol gland and trichome density can be best used as an index for screening large populations in sucki theng pest complex, and short list the advanced genotypes for evaluation at the field level [27,28].

## 5. Conclusions

This comparative study between *G. arboreum* and *G. hirsutum* cotton species demonstrated that *G. arboreum* genotypes exhibited a lower incidence of sucking pests compared to *G. hirsutum* genotypes. Timely pest control interventions are crucial to avoid yield penalties caused by alterations in chlorophyll content due to pest infestation. Negative significant correlations were observed between gossypol glands, trichome density, and other sucking pests such as aphids, Cicadellidae, and thrips. The inherent capacity of

the plant to produce higher quantities of gossypol glands and have enhanced trichome density appears to contribute to resistance against the sucking pest complex. Parameters such as chlorophyll content, gossypol gland quantity, and trichome density can serve as useful indices for screening large populations in the sucking pest complex and shortlisting advanced genotypes for field-level evaluation.

The identified genotypes can serve as parental lines in breeding programs to introduce desirable traits associated with tolerance to sucking pests. This approach can expedite the development of organic crop varieties with enhanced pest tolerance. In addition, farmers can cultivate these genotypes, contributing to the advancement of sustainable and environmentally friendly organical farming practices.

**Supplementary Materials:** The following supporting information can be downloaded at: https://www.mdpi.com/article/10.3390/agriculture13071402/s1, Table S1: Sucking pest data of the year 2019–2020 and 2020–21.

**Author Contributions:** Conceptualization, S.B.D., T.J. and A.R.; Methodology, S.B.D., M.M.M. and T.J.; Software, A.R.; Validation, S.B.D., A.N.P. and A.R.; Formal analysis, S.S.A. and S.B.D.; Investigation, S.S.A.; Resources, N.M.K.; Data curation, S.B.D. and T.J.; Writing—original draft, S.S.A.; Writing—review and editing, S.B.D., N.M.K., A.N.P., T.J., M.M.M. and A.R.; Visualization, T.J.; Supervision, S.B.D., N.M.K., T.J. and A.R.; Project administration, T.J. and A.R.; Funding acquisition, A.R. and M.M.M. All authors have read and agreed to the published version of the manuscript.

**Funding:** This research was funded by Mercator Foundation and Organic Cotton Accelerator grant number [65141] and the APC was funded by Mercator Foundation and Organic Cotton Accelerator grant number [65141].

**Institutional Review Board Statement:** Ethical review and approval was not required for the study on human participants in accordance with the local legislation and institutional requirements. The patients/participants provided their written informed consent to participate in this study.

**Data Availability Statement:** The raw data supporting the conclusions of this article will be made available by the authors, without undue reservation.

**Conflicts of Interest:** The authors declare no conflict of interest.

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
