# Peer review of "Studies on Morphophysiological and Biochemical Parameters for Sucking Pest Tolerance in Organic Cotton"

_agriculture, doi:10.3390/agriculture13071402_

Round 1
Reviewer 1 Report
Major comments:
M&M: The information of twenty cotton genotypes is incomplete. What is the full name of SGF? Which cultivated species of these genotypes is belong to?
Results: the data of sucking pest population was a lack of statistical analysis, it is difficult to reach a convincing conclusion.
Minor comments:
Line 86: What is the full name of DAS?
Fig. 1 and Fig 4: Missing notes.
Reviewer 2 Report
Studies on morphophysiological and biochemical parameters for sucking pest tolerance in organic cotton have been reviewed
I recommend including the abstract, general description of methods, treatments, or evaluations, main results expressed with values and statistical significance, and the conclusion of the evaluation or analysis of the experimental results.
Please add the hypnosis at the end of introduction
The discussion is unsuitable to publish, you must focus on your work by discussing your results step by step and some of the citations remove from the discussion is suitable to mention in the section of introduction. The conclusions are weak.
All tables must be self-explanatory.
Please cross match all the references in the text and in the list and vise versa.
This paper can be accepted after Major revisions
Minor editing of English language required
Reviewer 3 Report
Manuscript title: Studies on morphophysiological and biochemical parameters for sucking pest tolerance in organic cotton
General comments:
The manuscript reports a study on different species and varieties important for cotton production. The authors studied pest incidence along with several features of the plants. Overall the authors gathered valuable data, however the evaulation and presentation of the data must be improved.
The manuscript should be more conclusively structured, for instance results should not be referred to in Material and Methods section.
The authors did not compare different variables statistically, this must be done and Discussion should be reconsidered accordingly.
Only correlations were calculated, however, no details are given on the method used.
Although there were replicates, results are indicated by one number, possibly the mean, without standard deviation or standard error, this should also be corrected both in tables and throughout the text.
Please add a list of tested species and varieties, to make the setup clear to the readers.
Results should be presented more concisely, Table 1 should be given as Supplementary material.
For several figures no legends were given, this should be corrected.
Please list which pest species were caught, if the material was only determined to larger taxonomic groups, state it in Material and Methods.
Instead of ’jassid’ please use ’cicadellid’ throughout the text.
In figures 6-9: the blue bars of 30 and 90 DAS look rather similar, please use more contrasting colours.
The language of the text is in general good, however, some minor language editing could further improve it.
Specific comments:
Lines 21-22: Please indicate the country
Line 38: Malvaceae
Lines 36-41: please add reference
Line 47: for human health
Line 48: please rephrase the sentence
Line 56: instead of ’protecting’ please use ’lesser harm to’
Lines 53-55, 58-59: These sentences call for social activity and do not add to the scientific background of the study. Please rephrase these sentences.
Lines 61-62: please add reference to the statements
Line 73: please explain the Kharif season
Lines 80,82: in both years
Line 86: Please explain DAS
Lines 88-117: In my opinion the subheadings are not necessary here
Line 127: were kept in hot water bath
Line 135: absorbances were also recorded
Line 171: aphid population
Table 3: no details on the correlation method and meaning of asterisks are given
The language of the text is in general good, however, some minor language editing could further improve it.
Round 2
Reviewer 1 Report
Thanks to the authors for their edits to this manuscript. I have no more further comment.
Author Response
Thank you for reviewing the revised manuscript. We appreciate your time and effort in providing constructive feedback. Your comments have helped us to revive the manuscript and made it more reader friendly. Thanks again.
Reviewer 2 Report
I recommend accepting this manuscript in the present form
Minor editing of English language required
Author Response
Thank you for your recommendation to accept the manuscript in its present form with minor editing of the English language. We have made best possible efforts to ensure that the manuscript undergoes a thorough language editing process to address any minor language issues and improve its overall readability. We appreciate your support and look forward to the publication of the manuscript.